# A Novel Smart Belt for Anxiety Detection, Classification, and Reduction Using IIoMT on Students’ Cardiac Signal and MSY

**DOI:** 10.3390/bioengineering9120793

**Published:** 2022-12-12

**Authors:** Rishi Pal, Deepak Adhikari, Md Belal Bin Heyat, Bishal Guragai, Vivian Lipari, Julien Brito Ballester, Isabel De la Torre Díez, Zia Abbas, Dakun Lai

**Affiliations:** 1School of Life Science and Technology, University of Electronic Science and Technology of China, Chengdu 610056, China; 2School of Information and Software Engineering, University of Electronic Science and Technology of China, Chengdu 610056, China; 3IoT Research Center, College of Computer Science and Software Engineering, Shenzhen University, Shenzhen 518060, China; 4Centre for VLSI and Embedded System Technologies, International Institute of Information Technology, Hyderabad 500032, India; 5Department of Science and Engineering, Novel Global Community Educational Foundation, Hebersham, NSW 2770, Australia; 6Research Group on Foods, Nutritional Biochemistry and Health Universidad Europea Del Atlántico, Isabel Torres, 39011 Santander, Spain; 7Research Group on Foods, Nutritional Biochemistry and Health Universidade Internacional do Cuanza, Cuito EN250, Angola; 8Research Group on Foods, Nutritional Biochemistry and Health Universidad Internacional Iberoamericana, Arecibo, PR 00613, USA; 9Department of Signal Theory and Communications and Telematic Engineering, University of Valladolid, Paseo de Belén 15, 47011 Valladolid, Spain; 10School of Electronic Science and Engineering, University of Electronic Science and Technology of China, Chengdu 610056, China

**Keywords:** yoga, anxiety, machine learning, internet of medical things, student, artificial intelligence, therapy, integrative medicine, exercise, health, brain

## Abstract

The prevalence of anxiety among university students is increasing, resulting in the negative impact on their academic and social (behavioral and emotional) development. In order for students to have competitive academic performance, the cognitive function should be strengthened by detecting and handling anxiety. Over a period of 6 weeks, this study examined how to detect anxiety and how Mano Shakti Yoga (MSY) helps reduce anxiety. Relying on cardiac signals, this study follows an integrated detection-estimation-reduction framework for anxiety using the Intelligent Internet of Medical Things (IIoMT) and MSY. IIoMT is the integration of Internet of Medical Things (wearable smart belt) and machine learning algorithms (Decision Tree (DT), Random Forest (RF), and AdaBoost (AB)). Sixty-six eligible students were selected as experiencing anxiety detected based on the results of self-rating anxiety scale (SAS) questionnaire and a smart belt. Then, the students were divided randomly into two groups: experimental and control. The experimental group followed an MSY intervention for one hour twice a week, while the control group followed their own daily routine. Machine learning algorithms are used to analyze the data obtained from the smart belt. MSY is an alternative improvement for the immune system that helps reduce anxiety. All the results illustrate that the experimental group reduced anxiety with a significant (*p* < 0.05) difference in group × time interaction compared to the control group. The intelligent techniques achieved maximum accuracy of 80% on using RF algorithm. Thus, students can practice MSY and concentrate on their objectives by improving their intelligence, attention, and memory.

## 1. Introduction

University populations suffer from higher rates of mental health problems rates than societies before the pandemic, and the most common mental symptoms are depression, anxiety, and stress [1,2,3]. During the university age, both the environmental and developmental factors are changing at a crucial time [4]. As university students move from adolescence to adulthood, they experience such transitions as finding their identity, adjusting to societal values, and reaching maturity physically and socially [1]. According to Liu et al. [5], university students are more likely to experience depression, anxiety, and stress in the first year. Stress can develop when students adjust to their new university environment, live away from family, get used to living in a dormitory, and become economically independent [6]. Similarly, academic pressure also leads to stress in students. The cumulative grade point average (GPA) from the previous semester or year is used to measure academic performance. Low academic achievement may prevent students from enrolling in reputable companies or good universities for higher education degrees [1,7].

A healthy body necessitates a robust immune system and a healthy mind, all of which can be achieved by regular exercise involving balance, stretching, flexibility, strength training, and deep breathing. Yoga is a conscious process of gaining control over the mind, which is a twofold procedure: (i) to gain a capacity to focus and concern, (ii) to learn effectively to calm down the mind. Hence, it is considered an excellent form of exercise that improves people’s mental and physical health by stretching numerous body organs associated with breathing techniques. There are many different types of yoga postures, which can be divided into four categories: sitting, standing, prone, and lying. All of these positions require a breathing mechanism, which includes proper inhalation and exhalation. Physical posture, also known as *asanas*, breathing technique known as *pranayama*, and meditation known as *dhanaya*, are the three important aspects of yoga, which help to bring all-around personality development, including physical, mental, intellectual, emotional, and spiritual. Apart from that, yoga possesses an inimitable calming and peaceful sensation [8], useful in numerous aspects of well-being, including anxiety and stress management, sleep, diabetes, cardiovascular illnesses, cancer, and body balance [9,10,11]. Recently, yoga has risen in popularity among university students for handling anxiety and stress [1,2,7,12]. However, incorrect ways of practicing yoga can cause significant injuries such as strokes and other physical risks. To address this issue, numerous wearable devices [13], intelligent yoga mats [14], and yoga detection sensors [15] have been designed to detect and monitor human health and yoga positions.

The healthcare industry has undergone radical transformations with the integration of artificial intelligence and Internet of Medical Things (IoMT) as the Intelligent Internet of Medical Things (IIoMT), recalibrating a plethora of applications within the framework [16,17]. Healthcare 4.0, the recent generation, enhances health services by allowing and alerting medical practitioners in a timely manner to monitor the health information shared through multiple sensors, including wearable devices. The biosensors integrated with intelligence approach such as deep learning can alert medical practitioners and hospitals to be more vigilant during irregular occurrences [18,19,20]. To reap the full benefits of yoga, proper posture and techniques are required, which necessitates the observation and supervision of yoga professionals.

Shohani et al. [21] studied the impact of Hatha yoga on stress, anxiety, depression in women, using Depression Anxiety Stress Scale-21 (DASS-21) for the analysis of the data obtained from the questionnaire. Simon et al. [22] implemented Kundalini yoga and cognitive behavior therapy for the reduction of generalized anxiety disorder. A cross-sectional study was performed by Sahni et al. [23] on the self-management of the stress-related problem by using yoga for COVID outbreak subjects. Data were collected by designing the Brief Illness questionnaires through an online survey. DASS-9 was used for the measurement of stress, anxiety, and depression. All the analysis was performed based on the data from the online survey. Rachakonda et al. [24] designed SaYoPillow, smart yoga pillow for stress management by taking into consideration sleeping habits. The proposed framework relies on the blockchain-integrated privacy-assured IoMT. Bressington et al. [25] proposed the practicality and possible efficacy of full-scale randomized controlled trail (RCT) of laughter yoga for reducing residual mood, anxiety, and stress symptoms in adults who had been diagnosed with depression. The participants completed baseline (T0), Client Satisfaction Questionnaire at post-intervention (T1), and qualitative interviews at follow-up session (T2). Tolahunase et al. [26] studied a variety of indicators (biomarkers) that show neuroplasticity in subjects with depression through a 12-week yoga and meditation course. The biomarkers included brain-derived neurotrophic factor (BDNF), DNA damage, oxidative stress, and telemore metabolism. Improvement in systemic neuroplasticity biomarkers, in conjunction with a favorable clinical result showed that yoga with meditation may provide long-term clinical therapy to depressive subjects.

The main objective of the paper is to detect, estimate, and reduce anxiety using wearable sensor and SAS, machine learning, and MSY. The detection of the anxiety is accomplished by using SAS, an online questionnaire system [27], and smart belt. The system assigns the score based on the defined protocols. Then, the smart belt is used to detect the heart rate for the detection of anxiety. Machine learning algorithms are used to analyze the data obtained from the smart belt. Finally, MSY is performed as a measure to reduce anxiety.

The fundamental contribution of this research paper are:Anxiety is detected and monitored using a wearable smart belt and SAS. Bio-signals obtained from the chest-based sensors were implemented to generate high-level feature descriptions and generalization capabilities;The obtained signal undergoes various processing and analysis tasks using a machine learning approach, such as random forest, decision tree, and AdaBoost algorithms;Yoga is used as a tool to reduce anxiety by following a special protocol designed;Pre and post-yoga performance are taken for analysis through wearable smart belt data, and the Self-rating Anxiety Scale (SAS);All the results obtained illustrate that the MSY is able to reduce stress.

The rest of the article is structured as follows. In Section 2, the detailed experimental data collection is discussed. The methods are evaluated in Section 3. Section 4 discusses the analysis and results. Finally, Section 5 concludes this article by summarizing the achievements and stating possible future directions.

## 2. Experimental Data Collection

### 2.1. Smart Belt

Various reasons lead to the stressful lives of human beings deteriorating healthy living. To maintain a healthy living standard, detecting and monitoring anxiety is essential, which is a challenging task. With the objective to monitor and detect the variation of heart rate of the participants having anxiety, we used an intelligent belt designed and developed by Hexin Median Co. Ltd. in Shenzhen, China [28]. The smart belt implements an ECG recorder consisting of silver-coated dry electrodes, screen-printing technology, a pre-amplifier, microcontroller, an analog-to-digital conversion, and uses Bluetooth to send the received signals to a smartphone [29]. The silver-coated dry electrodes are pasted on the left and right sides of the chest, which are connected to the recorder base. The device can record ECG signals on a real-time basis for more than 24 h. The sampling frequency of the ECG signal is 250 Hz. Figure 1 shows the smart belt that we implemented in our research for data collection. The data collected from the sensor were forwarded to the mobile through Bluetooth and then forwarded to the server for further processing.

### 2.2. Participants

Out of 150 subjects registered for the MSY training program at the University of Electronic Science and Technology of China (UESTC) Qingshuihe campus Chengdu, China, 77 participants with Anxiety traits covered within inclusive criteria were screened out. Secondly, out of 77 eligible subjects, 66 subjects were selected and recruited on the basis of voluntary willingness to give informed consent. They were randomized into two groups on a lottery basis comprising volunteers, one experimental and the other control group. Consequently, the number of participants in the experimental group was fixed at 39 and that of the control was at 27. On the post data collection 12 and 10 dropouts were seen in the experimental and control group, respectively, as shown in Figure 2. Those students not participating in final data collection were considered as dropouts. The ethical committee of the University of Electronic Science and Technology of China (UESTC), Sichuan, China approved this study, and written consent was obtained for all participants (Ethics acceptance number 1061420210829008, obtained on 26 December 2021). This study was conducted on the premises of the UESTC after obtaining permission.

### 2.3. Study Design and Procedure

All the students detected with anxiety, from results obtained from the questionnaire and smart belt, were considered a high-risk group and were divided into two groups: anxiety and the control group. The experimental group is also termed as yoga group. Those who were not detected as having anxiety were excluded from further experiments, as they belonged to the healthy group. The anxiety group, also termed as experimental group, followed a yoga intervention for one hour twice a week for six weeks. The intervention module designed for the experimental group followed a novel yoga well design series called MSY comprising *Surya Namaskara, asana, pranayama, kriya*, meditation, and relaxation. The training module had 12 sessions in 6 weeks. MSY series is practiced as it is in all sessions, and the list of all practices is provided. A recommended integrated MSY module prepared for the Anxiety trait with various components including breathing practices, loosening exercises, Surya namaskara, postures (Asana), cleansing processes (Kriya), meditation (Dhyana), and relaxation was allowed to practice by the Yoga group. The detail of the MSY module followed in the present study is shown in Table 1. Similarly, the control group followed day-to-day life as it is and did not perform any type of assigned exercise or yoga training during the time period.

### 2.4. Psychometric Scales

Psychometric questionnaires based on the Self-rating Anxiety Scale (SAS) were developed primarily as a measure of somatic symptoms associated with the anxiety responses by W.K. Zung [27]. The SAS is an online questionnaire system, which the participants accessed, and the system gave the score according to the defined protocols [30]. The SAS scale measures competitive anxiety as a trait rather than an anxious state [31,32]. Thus, anxiety evaluated by the SAS scale reflects a rather steady and consistent tendency to feel anxious in scenarios leading up to and during academic competition. For instance, during academic competition, anxiousness as a state might relate to a momentary psychological and physiological reaction. Those scores were used to analyze the anxiety level of the participants [33]. The SAS ranges from a raw score of 20 to 80. According to the results of Chinese norm, the cut-off value of SAS standard score is 50 points, of which 50–59 points represent mild anxiety, 60–69 points are moderate anxiety, and more than 70 points are severe anxiety. The more anxious subject are, the higher score they obtain on the scale. The SAS is a 20-item self-report scale designed to evaluate a person’s anxiety level that covers a wide range of anxiety symptoms, both somatic (for example, “I feel my heart beating faster”, “My arms and legs shake and tremble”) and psychological (for example, “I feel like I’m falling apart and going to pieces”, “I feel afraid for no reason at all”) in naturewell-designs are given on a 4-point scale ranging from 1 (none, or only a little of the time) to 4 (a lot of the time). Participants were instructed to provide their responses on their previous week’s experience. Both negative and positive experiences (such as, “I fall asleep quickly and have a nice night’s sleep.”) were also included, where the latter experience were reverse-scored [27]. According to the China Mental Health Center, the normal range is 25–49, mild anxiety is 50–59, moderate anxiety is 60–69, and severe anxiety is ≥70.

## 3. Methods

### 3.1. Assessment Schedule

With the aim to assess the impact of yoga on anxiety disorder participants, various psychological characteristics and clinical parameters, i.e., psychological (SAS) and Heart Rate Variables (HRV) were sampled. Demographic and anthropometric details of the participants, viz., Body Mass Index (BMI) at the baseline were measured in the Mental health department of UESTC. Participants from the experimental group were asked to collect the ECG samples before and after the training period. ECG data were collected for the time duration of 20 min in both starting and ending times after undergoing a relaxation period for participants.

### 3.2. Electrocardiogram

Electrocardiogram (ECG) is the primary medical diagnostic tool for disease in practice and it provides a comprehensive picture of a patient’s cardiac conditions [34,35]. Currently, physicians usually use post hoc analysis through ECG waveforms to diagnose whether a patient is well or sick, which is inefficient, time-consuming, and unreliable due to physicians’ experience and expertise level [36]. Computer-aided automatic ECG analysis could effectively enhance the diagnosis efficiency as well as shorten diagnosis time [19,37]. The HRV analysis was conducted for the ECG signals in the time domain and frequency domain using PhysioNet Cardiovascular Signal Toolbox, MATLAB which was created as a part of the Vest et al. [38] research on open-source HRV analysis tool.

### 3.3. Minimum Redundancy Maximum Relevance (mRMR)

mRMR selects features based on maximum relevance and minimum redundancy criteria. Based on reciprocal knowledge, the maximum relevance criterion selects the most closely related features to the label [39,40]. The mutual information for the variables *a* and *b* can be written as shown in Equation (Equation 1): (1)$$I\left(a,b\right)=\sum_{a,b}p\left(a,b\right)log\left(\frac{p\left(a,b\right)}{p\left(a\right)p\left(b\right)}\right)$$ Maximum relevance $M_{r}$ for a feature *f* can be calculated as
(2)$$M_{r}=I\left(f,l\right)$$
where $p\left(a,b\right)$ is the joint probabilistic density, $p\left(a\right)$ and $p\left(b\right)$ are marginal probabilistic densities, *l* represents the target level. The feature with maximum relevance is ranked at position one and the feature with the lower D value, then the first feature is ranked at position two, and so on. The lowest redundancy criterion chooses the characteristic with the least redundancy among those chosen by the highest relevance criterion. The redundancy of features can be calculated as follows:(3)$$R=\frac{1}{\left|L\right|}\sum_{fIL}I\left(f,f^{{}^{\prime }}\right)$$
where *L* is the set of features with the lowest redundancy level and is the most relevant to the goal label.

### 3.4. Analysis of Variance (ANOVA)

When employing ANOVA, the F-value may be utilized to choose attributes. The variance ratio between features and within samples is the F-value. The stronger a characteristic’s F-value, the better it can differentiate between positive and negative samples [41]. The F-value for a feature *m* is calculated as:(4)$$F\left(m\right)=\frac{s_{B}^{2}\left(m\right)}{s_{W}^{2}\left(m\right)}$$
where $s_{B}^{2}\left(m\right)$ is the variance between the features and $s_{W}^{2}\left(m\right)$ is the variance within the samples of each feature. These variances can be represented as
(5)$$s_{B}^{2}\left(m\right)=\frac{\sum_{i=1}^{k}n_{i}{\left(\frac{\sum_{i=1}^{n_{i}}f_{ij}\left(m\right)}{n_{i}}-\frac{\sum_{i=1}^{k}\sum_{j=1}^{n_{i}}f_{ij}\left(m\right)}{\sum_{i=1}^{k}n=i}\right)}^{2}}{df_{B}}$$
(6)$$s_{B}^{2}\left(m\right)=\frac{\sum_{i=1}^{k}\sum_{i=1}^{n_{i}}{\left(f_{ij}\left(m\right)-\frac{\sum_{i=1}^{k}\sum_{j=1}^{n_{i}}f_{ij}\left(m\right)}{\sum_{i=1}^{k}n_{i}}\right)}^{2}}{df_{W}}$$
where *K* denotes the total features, *N* denotes the total samples, $f_{ij}\left(m\right)$ denotes *m*th feature of *j*th sample in *i*th group, $n_{i}$ denotes sample in *i*th group. The degree of freedom for between features $df_{B}$ and within samples $df_{W}$ was K−1 and N−1, respectively.

### 3.5. Intelligent Approach

For the classification of the data, we implement various intelligent approaches such as:

#### 3.5.1. Decision Tree (DT)

DT is a supervised machine learning model to predict the value of the target variables by a set of rules through the structure made as edges or branches, nodes, and leaf nodes [42,43]. Datasets are divided into leaves, where each leaf contains mutually exclusive records. DT can be used both for classification and regression problems [44,45]. Here, we use DT as a classification problem, where the classifier parameters include two minimum numbers of instances in leaves, a split subset that must be smaller than five, and a maximum tree depth of one hundred. The DT classifier has the following advantages: it is economical, quick, noise-free, and easier to implement than other classifiers [46]. DT is described in Equations (7) and (8):(7)$$E\left(H_{t}\right)=\sum_{j}P_{j}H_{j}$$
(8)$$R_{t}=H-E\left(H_{t}\right)$$
where $H_{t}$ is the average uncertainty after performing test *t*, $P_{j}$ is the probability that the test has *j* outcome, and $R_{t}$ is the average reduction in uncertainty achieved by test *t*.

#### 3.5.2. Random Forest (RF)

RF, also called random decision forests, is an ensemble learning method for regression, classification, and other tasks that operates by constructing a multitude of decision trees at training time [42,47]. For classification, the RF output is the class selected by the most trees, and regression is the mean or average prediction of the individual predictions returned [18]. Each individual tree consists of a decision node, a leaf node, and a root node, where output is the leaf node, and a majority voting mechanism determines the final result [45,47]. If we have attributes Θ of a vector *x* and decision tree based on these attributes is h(x,Θ), then the random forest can be defined as
(9)$$f=h\left(x,\Theta_{k}\right),$$
where, k=1,2,…k [48].

#### 3.5.3. Adaboost Classifier

Adaboost classifier, an ensemble learning method, also called meta-learning, is an iterative approach that incorporates learning from weak classifiers and turning them into strong ones [49,50,51]. In each iteration, the data samples are trained based on various weights of the classifier to minimize the training errors that can assure accurate predictions. Adaboost must adhere to the below requirements [52,53]:On a variety of weighed training instances, the classifier should be trained interactively;It strives to minimize training errors in order to offer a perfect fit for these examples in each iteration.

### 3.6. Evaluation Measures

For the deep analysis of the model, evaluation measures are essential. Therefore, we considered four significant evaluation measures to capture different perspectives of outcomes. The chosen measures quantify the effectiveness of the classification model [54,55,56]. Here, we have discussed the basic knowledge to understand how various performance metrics such as True Positive (TP), True Negative (TN), False Positive (FP), False Negative (FN), sensitivity, and specificity, are calculated.

Considering the example of a medical test for diagnosing a condition.

*True Positive (TP):* Diseased people correctly tested as diseased. Examples of training where the true class is positive and we hypothesized as positive. They are what is known as true positives, as shown in Table 2, represented by *a*.*False Positive (FP):* Healthy people incorrectly tested as diseased. This indicates that the examples are actually negative and that the learning algorithm is incorrectly classifying them as, as shown in Table 2, represented by *b*.*False Negative (FN):* Diseased people incorrectly tested as healthy. This indicates that the examples are in fact positive, but the learning algorithm incorrectly classifies them as negative, as shown in Table 2, represented by *c*.*True Negative (TN):* Healthy people correctly tested as healthy. Examples of training where the true class is negative and we hypothesized as Negative. They are referred to as true negatives, as shown in Table 2, represented by *d*.

Based on these, we compute and evaluate classifier performance considering Specificity ($S_{p}$), Sensitivity ($S_{n}$), Accuracy (Acc), Precision [44,52,57].

*Specificity (***$S_{p}$***):* Specificity refers to the test’s ability to correctly reject healthy patients without a condition. Mathematically, this can be expressed as:
(10)$$S_{p}=\frac{TN}{\left(TN+FP\right)}$$*Sensitivity (***$S_{n}$***):* Sensitivity refers to the test’s ability to correctly detect diseased patients who do have the condition. Mathematically, this can be expressed as:
(11)$$S_{n}=\frac{TP}{\left(TP+FN\right)}$$*Accuracy (***$A_{cc}$***):* Accuracy measures the complete rate of appropriate prediction.
(12)$$Acc=\frac{\left(TP+TN\right)}{\left(P+N\right)}=\frac{\left(TP+TN\right)}{\left(TP+TN+FP+FN\right)}$$*Precision:* The parameter that states the proportion of the subjects model marked as benign are truly benign.
(13)$$Precision=\frac{TP}{\left(TP+FP\right)}$$

## 4. Results and Discussion

Anxiety is a disease that affects people of different ages and can last longer, creating an environment for another disease too. Anxiety detection has been accomplished by using SAS questionnaires as a standard technique for a long time [31] rather than detecting changes in human physiology, such as heart rate signals, etc. So far, most of the existing systems are restricted for anxiety detection by observing its biological and biochemical effects, where various techniques, for example, blood pressure and heart rate are physiological effects that could be observed. However, biochemical, physiological, and biological-based methods have shown inconsistent findings because they depend on unstable hormones, depending on several factors such as gender, season, mood, age, well-being status, medicines, and smoking [58,59]. On the other hand, anxiety can be detected based on various bio-signals, e.g., EEG, ECG, electromyography, Respiration, Blood Volume Pulses, Blood Pressure, Skin Temperature, and Galvanic Skin Response [60,61,62]. Currently, with the growing popularity of the IIoMT, researcher and scientists are more focused on designing wearable sensors that can help to understand human behavior and physiology. In this way, anxiety disorders could be detected accurately and precisely through IIoMT in a reliable and timely manner.

### 4.1. Detection and Improvement of Anxiety

SAS scales were scored before and after 6 weeks of yoga training. The estimated marginal mean for the pre and post SAS is shown in Figure 3. Similarly, Figure 4 illustrates the results of the yoga and control group for both the pre and post time period. The results showed that after six weeks of yoga training (twice a week), the subjects’ SAS scores dropped significantly as shown in red color in Figure 4. SAS scores dropped significantly pre and post yoga (*p* < 0.0001), which suggests that anxiety and depression can be alleviated by 6 weeks of yoga training.

### 4.2. Classification of Anxiety

#### 4.2.1. Analysis and Feature Extraction

We used the cardiac (ECG) signal of the anxiety and control groups. We removed the noise of the signal using 200th-order low-pass FIR filter with 25 Hz cut-off frequency [63]. The missing signals were recovered using an imputation approach [42,64]. The most common method of HRV analysis, i.e., time domain analysis was applied to a sequence of successive normal inter-beat intervals. Then, the R waves, nearly a periodic occurrence of ECG signals, were detected using Pan–Tompkins technique [65]. Different features were extracted, such as NN50, the interval between two R-waves of duration more than 50 ms, pNN50, the interval between two R-waves whose proportion of neighboring NN intervals of duration more than 50 ms, RMSSD, the root mean square of adjacent differences of NN intervals and mean RR, mean value between two R-waves. We extracted the Heart Rate Variability of the ECG signal such as RMS, mean-RR, RMS SD, NN50, SD RR, PNN50, mean-ECG, and TPR, with demographic features such as Age (year), Height (cm), Weight (Kg), Body Mass Index (BMI).

Finally, one-minute ECG signal representation of the anxiety psycho-neurological human behavior is shown in Figure 5, where raw signal in Figure 5a, filter signal obtained from 200th-order low-pass FIR in Figure 5b, and R-R interval of the ECG signal in Figure 5c are presented. We used mRMR techniques for the feature selection. It showed that features age, RMS, NN50, TPR, SAS score, SD RR, and weight are significant. However, we used these features as an input for the classifiers. Additionally, the heatmap showed the relationship between features mentioned in Figure 6.

#### 4.2.2. Classifications (Internal and External) Results

The obtained data were classified as internal (pre vs. post) and external (control vs. yoga) in terms of AUC, accuracy, precision, sensitivity, and specificity. Additionally, we used DT, RF, and AB classifiers with leave-one-out and Cross-Validation (CV) with 3- and 5-fold termed as CV-3 and CV-5 models, respectively, as shown in Table 3.

In the internal (pre vs. post) classification, CV-3 model of the AB classifier achieved maximum performance measures in terms of AUC (0.75), accuracy (0.75), precision (0.75), sensitivity (0.75), and specificity (0.75). Additionally, CV-5 model of the DT classifier achieved the minimum performance measures in terms of AUC (0.68), accuracy (0.59), precision (0.59), sensitivity (0.59), and specificity (0.59).

In external (yoga vs. control) classification, the leave-one-out model of the RF classifier achieved maximum performance measures in terms of AUC (0.82), accuracy (0.80), precision (0.80), sensitivity (0.80), and specificity (0.73). Additionally, the CV-3 and CV-5 models of the AB classifier achieved the minimum performance measures in terms of AUC (0.69), accuracy (0.72), precision (0.72), sensitivity (0.72), and specificity (0.67).

The first goal of this research is to observe the immediate consequences of yoga practice on changes in SAS level and ECG parameter which in turn reflect participants’ expectation of reducing stress and depression. The SAS level and ECG results showed improved performance more in response to a yoga session than to a control group. In support of this claim, a previous study on health subjects found that a subtle instruction to minimize anxieties and increase relaxation the next day was associated with substantially lower levels [66]. Physical movement, meditation, and yogic breathing are some of the elements of the integrated yoga intervention that have been shown to elicit the relaxation response [67]. Yoga postures, breathing techniques, and their interconnections with mindfulness-based therapies have dramatically enhanced sleep efficiency caused by anxiety-related problems, and cancer sufferers [68]. Previous research has demonstrated that yoga is an effective stress management approach [2].

### 4.3. Comparison between Previous and Proposed Study

We compare the proposed method with the various existing literature [37,69,70,71] in terms of AUC, sensitivity, specificity, and accuracy, as presented in Table 4. The proposed method performs better (accuracy: 80%) than the compared techniques in all the performance indicators. This proves the efficacy of the proposed method in the detection, estimation, and reduction of anxiety. Previously, Nemesure et al. [37] approached the problem of predicting anxiety using a machine learning pipeline to re-analyze data from an observational study. They obtained that model performance in terms of AUC is 0.73 for anxiety. Bokma et al. [69] predicted recovery from anxiety disorders within 2 years by applying a machine learning approach. His RF model performance in terms of AUC is 0.67 for predicting recovery from anxiety disorders. Júnior et al. [70] designed a model SVM that achieved performance in terms of AUC is 0.82 for the detection of anxiety. However, our models DT and RF are better than others (Figure 7). The major reasons behind obtaining better results include: we used a flexible electronics-based wearable smart belt that helped us to collect the biological data (ECG signal) in a scientific and significant manner, containing less noise and low error. The mRMR approach delivered better significant values on Demographic and ECG extracted features. Those significant features were used as input in machine learning models, which helped to achieve better results. Similarly, we applied ANOVA on SAS-collected scores to find the improvements of life through MSY on anxiety. However, we found our novel MSY is more effective in reducing students’ anxiety. All the results illustrated above present that MSY has a moderate effect on decreasing the level of anxiety disorder. As a result, MSY may be a promising trans-diagnostic strategy for the reduction of anxiety disorder symptoms in persons suffering from an anxiety disorder or mental illness. MSY gives rise to significant healing effects through synchronizing and integrating mind and body. It could effectively induce mindfulness and self-compassion which are proven to be meditating factors between yoga and stress.

### 4.4. Applications and Limitations of the Study

The proposed work illustrated an application for detecting, estimating, and reducing anxiety, where wearable sensors and SAS questionnaires were used for detecting the anxiety, machine learning algorithms for estimating the results obtained from sensors and SAS questionnaires, and yoga is used for reducing the anxiety. This work provides a more effective and accurate detection and reduction of anxiety using MSY, where intelligent approaches are used for detection and estimation. The most important application of the present research is to diagnose anxiety subjects using wearable sensors and practice yoga for healthy living.

The present work has some limitations, including the small number of participants and HRV being used to detect anxiety. Further work could be required with a large number of participants with different age groups and implementing a deep learning approach for analysis. Similarly, other wearable devices that can detect EEG signals, blood pressure, etc., could be used to detect anxiety. In the future, we will be addressing all the limitations that occurred in this research.

## 5. Conclusions

This paper proposed a novel integrated detection-estimation-reduction framework for anxiety using the IIoMT and MSY. IIoMT is the integration of IoMT (wearable smart belt) and machine learning algorithms (DT, RF, and AB). Anxiety is detected using a wearable smart belt, and SAS, which was accomplished in the pre and post-yoga phases. This research discovered evidence of MSY having a favorable effect beyond standard care in lowering anxiety symptoms in persons with an anxiety disorder. The number of yoga sessions twice a week for six weeks showed a moderate improvement in anxiety disorder symptoms. Given the good findings of this study, MSY should be considered as an evidence-based exercise modality alongside traditional types of exercise. MSY can be an additional or alternative method for getting people who have anxiety to participate in meaningful physical activity. We would perform more experiments using multiple wearable sensors to detect anxiety and implement deep learning approaches in the future.

## Figures and Tables

**Figure 1 bioengineering-09-00793-f001:**
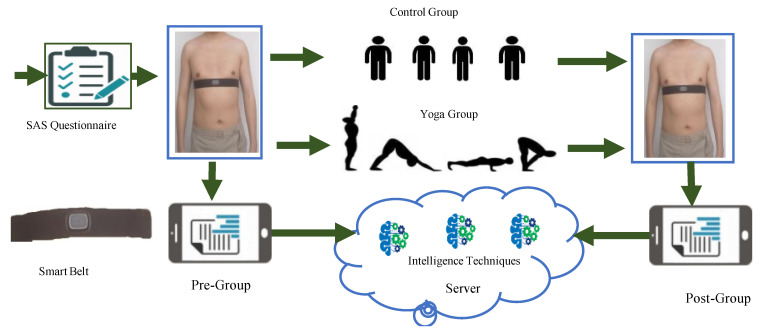
The workflow of the study to detect, estimate and reduce anxiety using yoga, IIoMT, and machine learning.

**Figure 2 bioengineering-09-00793-f002:**
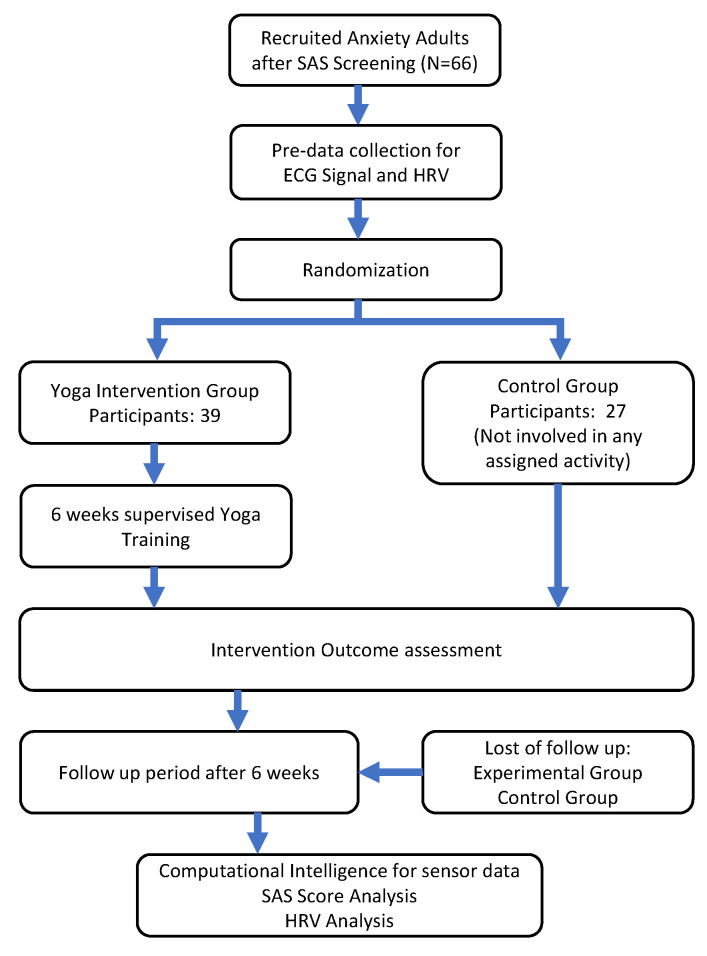
The flowchart for the theranostic approach of the anxiety neurological human behavior using MSY.

**Figure 3 bioengineering-09-00793-f003:**
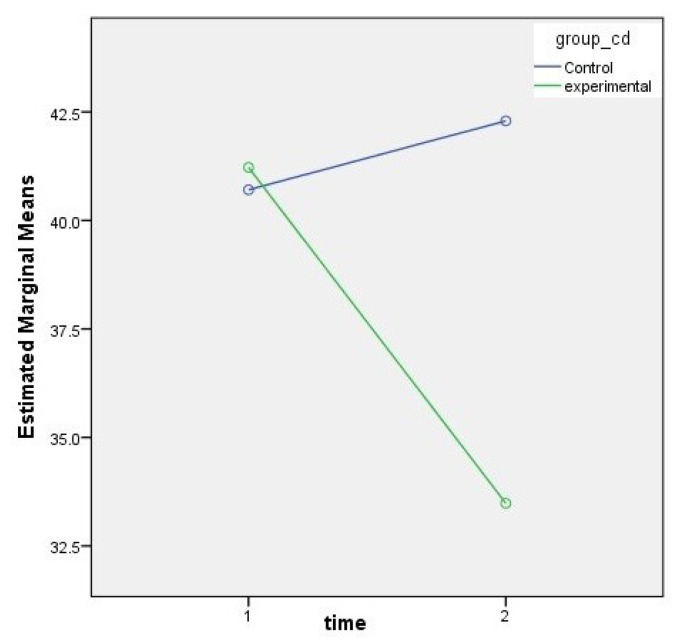
The estimated marginal mean obtained from data obtained from SAS questionnaire. Time 1 and 2 represents pre and post SAS results, respectively.

**Figure 4 bioengineering-09-00793-f004:**
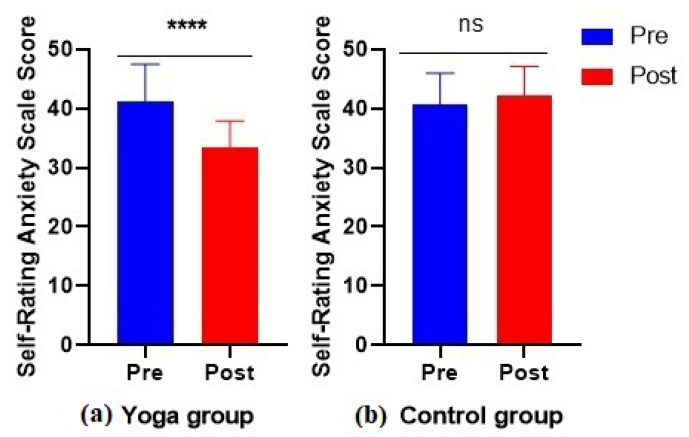
Results from the SAS questionnaire for two different groups (**a**) experimental group and (**b**) control group, where **** and *ns* represents significiant and non-significant respectively. Both groups were further divided into pre and post experimental and control group, respectively. Red and blue color represent pre and post experimental and control groups, respectively.

**Figure 5 bioengineering-09-00793-f005:**
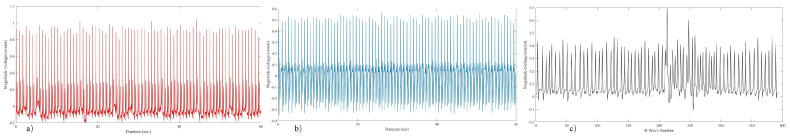
Sixty seconds ECG signal representation of the anxiety-suffering student including (**a**) raw signal, (**b**) filtered signal, and (**c**) R-R interval.

**Figure 6 bioengineering-09-00793-f006:**
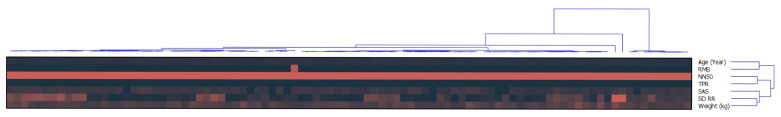
/hlHeat map based on mRMR-approved features.

**Figure 7 bioengineering-09-00793-f007:**
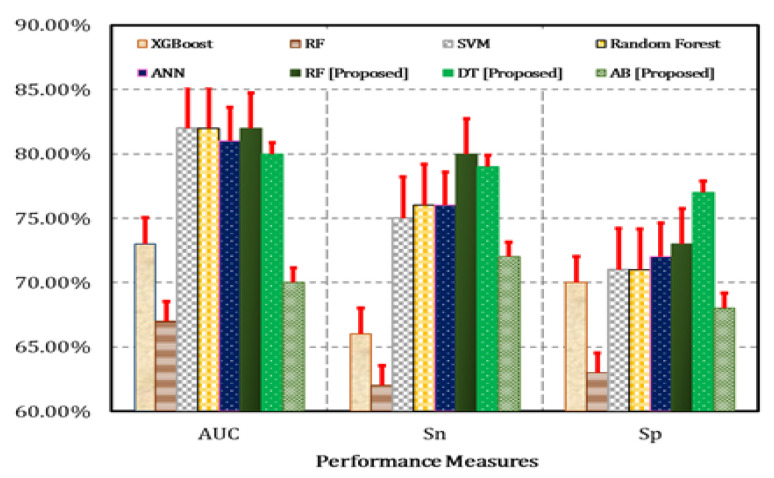
Comparison of the proposed approach with the state of the art [37,69,70,71,72].

**Table 1 bioengineering-09-00793-t001:** Mano Shakti Yoga (MSY) protocol designed for training session to handle anxiety.

Headings	Steps	Sanskrit Name	English Name	Repeat	Hold Time
Beginning	B_1	Vajrasana	Thunderbolt pose	Sitting slient	3 Min.
	B_2	Pada shakti	Ankle stretch	10 Rounds (R,L)	Last round 15 Sec.
	B_3	Kati Vakra	Back twisting	10 Rounds (R,L)	Last round 15 Sec.
	B_4	Kati chakra	Side stretch	10 Rounds (R,L)	Last round 15 Sec.
Sun Salutation	SS_1	Samastithi	Stand Straight	3	15 Sec.
	SS_2	Ardha chakra	Half moon pose	3	15 Sec.
	SS_3	Padahastasna	Hand to feet	3	15 Sec.
	SS_4	Ashwachalan	Horse ride pose (Rt leg Back)	3	15 Sec.
	SS_5	Chaturangasana	Plank pose	3	15 Sec.
	SS_6	Sashtanga namaskar	Eight limbs to floor	3	15 Sec.
	SS_7	Bhujangasana	Cobra pose	3	15 Sec.
	SS_8	Parvatasana	Mountain pose	3	15 Sec.
	SS_9	Ashwachalana	Horse riding pose (RtLeg Fwd)	3	15 Sec.
	SS_10	Padahastasana	Hands to feet	3	15 Sec.
	SS_11	Ardhachakra	Half Wheel	3	15 Sec.
	SS_12	Samastithi	Stand straight	3	15 Sec.
Standing	K_1	Ardhakati chakrasana	Half moon	1	R-L sides 30 Sec.
	K_2	Vrikshasana	Tree pose	1	R-L sides 30 Sec.
	K_3	Ardhachakrasana	Half wheel	1	20 Sec.
	K_4	Padahastasana	Hand to feet	1	20 Sec.
	K_5	Parsavakonasana	Side stretch	1	20 Sec.
	K_6	Padottanasana	Legs apart forward stretch	1	20 Sec.
Sitting	S_1	Vakrasana	Back twist	1	20 Sec.
	S_2	Ustrasana	Camel pose	1	20 Sec.
	S_3	Pachimottanasana	Back stretch	1	20 Sec.
	S_4	Purvottanasana	Forward stretch	1	20 Sec.
	S_5	Navasana	Boat pose	1	20 Sec.
Laying	L_1	Stimulate & Relax	Stimulation, relaxation tech	3 × 2	6 Min
	L_2	Savasana	Corpse pose	1	8 Min
Yogic Breathing	YB1	Bhramari	Humming breath	5 × 1	3 Min.
Meditation	M_1	Dhyana	Breath awareness	1	5 Min.
Ending	E_1	Vajrasana	Thunderbolt pose	1	3 Min.

**Table 2 bioengineering-09-00793-t002:** The confusion matrix for a binary classification model.

	Disease Present	Disease Absent
**Test Positive**	a (TP)	b (FP)
**Test Negative**	c (FN)	d (TN)

**Table 3 bioengineering-09-00793-t003:** Anxiety results of the various performance indicators for internal (pre vs. post) and External (control vs. yoga group).

		External (Control vs. Yoga)					Internal (Pre vs. Post)				
Model	Classifier	AUC	Acc.	Precision	$S_{n}$	$S_{p}$	AUC	Acc.	Precision	$S_{n}$	$S_{p}$
Leave One Out	DT	0.77	0.67	0.69	0.67	0.67	0.78	0.73	0.74	0.73	0.73
	RF	**0.82**	**0.80**	**0.80**	**0.80**	**0.73**	0.71	0.62	0.62	0.62	0.62
	AB	0.70	0.72	0.72	0.72	0.68	0.64	0.64	0.64	0.64	0.64
CV-3	DT	0.80	0.79	0.79	0.79	0.77	0.68	0.65	0.65	0.65	0.66
	RF	0.84	0.78	0.77	0.78	0.72	0.72	0.67	0.67	0.67	0.67
	AB	0.69	0.72	0.72	0.72	0.67	**0.75**	**0.75**	**0.75**	**0.75**	**0.75**
CV-5	DT	0.80	0.79	0.79	0.79	0.77	0.68	0.59	0.59	0.59	0.59
	RF	0.84	0.78	0.77	0.78	0.72	0.71	0.64	0.64	0.64	0.64
	AB	0.69	0.72	0.72	0.72	0.67	0.70	0.70	0.70	0.70	0.70
Mean		0.77	0.75	0.75	0.75	0.71	0.71	0.66	0.67	0.66	0.67
Standard Deviation		0.057	0.041	0.036	0.041	0.038	0.039	0.051	0.050	0.051	0.051
Variance		0.003	0.002	0.001	0.002	0.001	0.002	0.003	0.003	0.003	0.003

**Table 4 bioengineering-09-00793-t004:** Comparison between the proposed and existing models.

Ref.	Par.	Scoring Techniques	Classifier	Performance Measure			
				AUC	$S_{n}$	$S_{p}$	Acc.
[37]	4184	HADS, PHQ	XGBoost	0.73	0.66	0.7	
[69]	887	NESDA	RF	0.67	0.62	0.63	
[70]	200	PSWQ, HDRS, YMRS, CGI-S	SVM	0.82	0.75	0.71	
			RF	0.82	0.76	0.71	
			ANN	0.81	0.76	0.72	
[71]	154	STAI, MASQ-D, HDRS, HAM-A	Pattern regression				0.58
[72]	90	CAQ. HADS	RF				0.64
Proposed	66	SAS	RF	0.82	0.80	0.73	0.80
			DT	0.80	0.79	0.77	0.79
			AB	0.70	0.72	0.68	0.72

## Data Availability

Data are not publicly available due to privacy considerations. Data are available per request from Dr. Md Belal Bin Heyat (belalheyat@gmail.com).

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
