# Peer review of "A Novel Smart Belt for Anxiety Detection, Classification, and Reduction Using IIoMT on Students’ Cardiac Signal and MSY"

_bioengineering, 2022, doi:10.3390/bioengineering9120793_

Round 1

Reviewer 1 Report

In the reviewer’s opinion, the work here presented does look interesting but it needs extensive revisions before being accepted for publication. Specifically:

1.       Before coming to different aspects in need of revision, it must be stated that reading the body of the paper and the conclusion, it is not clear whether the aim of the authors is to show that the belt works or the yoga works. More precisely, it seems that authors are more interested to show a specific trend for a group of people and the capability to catch such a trend, more than showing the accuracy of the proposed procedure and protocol in comparison with possible alternatives in the scientific literature. Section 4.3 looks important, but no discussion is given: it is only said that some results are gathered in Table 3, and nothing more. This part needs instead to be expanded more, to provide a convincing comparison with other studies.

2.       Regarding the abstract and the discussion in the introduction, different aspects are not that clear. Is the period monitoring including also exam sessions? We mean, are the results almost constant in time, or do they have peaks in anxiety? People are said to be randomly selected, but nothing is said concerning what they might have in common and what distinguishes every single person in the cohort. Anxiety is declared on the basis of a self-test: how are results biased by this? How are people supported by psychologists or neuropsychiatrists to make sure that results are not actually biased by the aforementioned (maybe wrong) self-perception of anxiety? How can a baseline be set?

3.       Line 115: why mention a commercial product? Is this going to become a kind of advertisement for the company? Is it supporting the activities?

4.       Section 2.2: regarding ref. [27], it is not clear whether the surname of the author is Zuang or Zung. One single reference regarding the adopted protocol looks a bit too few to judge the efficacy of the proposed procedure. Please, enhance the discussion also in relation to the results obtained in the past with the same questionnaire. More details need to be added regarding the questions and “experiences”, rather than the yoga protocol.

5.       Regarding the participants, how is the cohort general enough to extend the results to any other possible group of students?

6.       Line 170: Table ??

7.       Section 3.5: variables within the text must be denoted with the same font as in the equations, and sentences after the equations starting with “where” do not have to have the letter W capitalized.

8.       ANOVA analysis: use the proper references here, since it is not new and the former works need to be given credit.

9.       Section 3.6: This section is written quite badly, probably not by the same person who wrote the other parts so far. Grammar is weaker, no explanations are given as paragraphs look like a bunch of sentences put together, and definitions of variables are not clear (even whether they are vectorial or scalar).

10.   Lines 223-224: we don’t get why anxiety detection is a hot investigation by “engineers”.

11.   Grammar must be checked accurately throughout.

Author Response

Reviewer 1

In the reviewer’s opinion, the work here presented does look interesting but it needs extensive revisions before being accepted for publication. Specifically:

  1. Before coming to different aspects in need of revision, it must be stated that reading the body of the paper and the conclusion, it is not clear whether the aim of the authors is to show that the belt works or the yoga works. More precisely, it seems that authors are more interested to show a specific trend for a group of people and the capability to catch such a trend, more than showing the accuracy of the proposed procedure and protocol in comparison with possible alternatives in the scientific literature. Section 4.3 looks important, but no discussion is given: it is only said that some results are gathered in Table 3, and nothing more. This part needs instead to be expanded more, to provide a convincing comparison with other studies.

Response: Thank you for the helpful suggestions. We have revised the abstract, body of the paper, and conclusion to present our objective in a clear and precise way. The objective of the paper is to show both the contribution of the paper: smart belt and MSY were able to achieve its goal as it was intended. MSY gives rise to significant healing effects through synchronizing and integrating mind and body. MSY  could effectively induce mindfulness and self-compassion which are proven to be meditating between yoga and stress. We compare the proposed method with the various existing literature [ 37 ,73 – 75 ] in terms of AUC, sensitivity, specificity, and accuracy as presented in Table 4. The proposed method performs better (accuracy: 80%) than the compared techniques in all the performance indicators. This proves the efficacy of the proposed method in the detection, estimation, and reduction of anxiety. Previously, Nemesure et al. [37 ] approached the problem of predicting anxiety using a machine learning pipeline to re-analyze data from an observational study. They obtained that model performance in terms of AUC is 0.73 for anxiety. Bokma et al. [73 ] predicted recovery from anxiety disorders within 2 years by applying a machine learning approach. His RF model performance in terms of AUC is 0.67 for predicting recovery from anxiety disorders. Júnior et al. [ 74] designed a model SVM that achieved performance in terms of AUC is 0.82 for the detection of anxiety. However our models DT and RF are better than others (Fig. 7). The major reasons behind obtaining better results include: we used a flexible electronics-based wearable smart belt that helped us to collect the biological data (ECG signal) in a scientific and significant manner, containing less noise and low error. The mRMR approach delivered better significant values on Demographic and ECG extracted features. Those significant features were used as input in machine learning models, which helped to achieve better results. Similarly, we applied ANOVA on SAS collected scores to find the improvements of life through MSY on anxiety. However, we found our novel MSY is more effective to reduce students’ anxiety. All the results illustrated above present that MSY has a moderate effect on decreasing the level of anxiety disorder. As a result, MSY may be a promising trans-diagnostic strategy for the reduction of anxiety disorder symptoms in persons suffering from an anxiety disorder or mental illness. MSY gives rise to significant healing effects through synchronizing and integrating mind and body. It could effectively induce mindfulness and self-compassion which are proven to be meditating between yoga and stress.

  1. Regarding the abstract and the discussion in the introduction, different aspects are not that clear. Is the period monitoring including also exam sessions? We mean, are the results almost constant in time, or do they have peaks in anxiety? People are said to be randomly selected, but nothing is said concerning what they might have in common and what distinguishes every single person in the cohort. Anxiety is declared on the basis of a self-test: how are results biased by this? How are people supported by psychologists or neuropsychiatrists to make sure that results are not actually biased by the aforementioned (maybe wrong) self-perception of anxiety? How can a baseline be set?

Response: Thank you for your sharp observations. We have rewritten the abstract.

The monitoring period is not included in exam sessions. Pressure to achieve good grades and age of 20-25 years are common in the randomly selected students. First anxiety is declared based on SAS results. To verify these results, we used a smart belt so that there would not be bias in the results. Aware of getting falsely detected anxiety (biased results), we analyzed the SAS  results by consulting the psychologist. It is to be noted that the results obtained by the smart belt were similar to SAS results. The SAS ranges from a raw score of 20 to  80. According to the results of Chinese norm, the cut-off value of SAS standard score is 50points, of which 50-59 points are mild anxiety, 60-69 points are moderate anxiety, and more than 70 points are severe anxiety. The higher anxious subject are, the higher score they get on the scale.

  1. Line 115: why mention a commercial product? Is this going to become a kind of advertisement for the company? Is it supporting the activities?

Response: We would like to thank the reviewer for the comments.

This is the new application for the smart belt, and we intend to provide the information that we used in the research. Similarly, we also found some papers that cited the company name, some ref. includes

  1. Achilli, A., Bonfiglio, A., & Pani, D. (2018). Design and characterization of screen-printed textile electrodes for ECG monitoring. IEEE Sensors Journal, 18(10), 4097-4107.
  2. Tsukada, Y. T., Tokita, M., Murata, H., Hirasawa, Y., Yodogawa, K., Iwasaki, Y. K., ... & Tsukada, S. (2019). Validation of wearable textile electrodes for ECG monitoring. Heart and vessels, 34(7), 1203-1211.

  1. Section 2.2: regarding ref. [27], it is not clear whether the surname of the author is Zuang or Zung. One single reference regarding the adopted protocol looks a bit too few to judge the efficacy of the proposed procedure. Please, enhance the discussion also in relation to the results obtained in the past with the same questionnaire. More details need to be added regarding the questions and “experiences”, rather than the yoga protocol.

Response: Thank you for your sharp observations. The typos have been corrected. And more related information and references have been added.

Psychometric questionnaires based on the Self-rating Anxiety Scale (SAS) was developed primarily as a measure of somatic symptoms associated with the anxiety responses by W.K. Zung [27 ]. The SAS is an online questionnaire system, which the participants accessed, and the system gave the score according to the defined protocols [ 30 ]. SAS scale measures competitive anxiety as a trait rather than an anxious state [31 ], [ 32]. Thus, anxiety evaluated by the SAS scale reflects a rather steady and consistent tendency to feel anxious in scenarios leading up to and during academic competition. For instance, during academic competition, anxiousness as a state might relate to a momentary psychological and physiological reaction. Those scores were used to analyze the anxiety level of the participants [33 ].

  1. Regarding the participants, how is the cohort general enough to extend the results to any other possible group of students?

Response: Thank you for your comments.

The proposed framework can be extended to other possible groups of students because of the following reasons:

  • MSY interventions helped to control the anxiety levels of all the participants from the anxiety group.
  • The anxiety group performed MSY one hour twice a week, which also helps to claim that when the MSY is practiced on a regular basis helps to control anxiety.
  1. Line 170: Table ??

Response: Thank you for your sharp observations. In the revised manuscript, we have corrected the typo, it is Table 1.

  1. Section 3.5: variables within the text must be denoted with the same font as in the equations, and sentences after the equations starting with “where” do not have to have the letter W capitalized.

Response: Thank you for your  kind advice. The typos, spelling, and grammatical errors have now been addressed by authors and professional proof-reader in the revised version.

  1. ANOVA analysis: use the proper references here, since it is not new and the former works need to be given credit.

Response: We agree with the reviewer. We have cited the paper with proper references.

  1. Section 3.6: This section is written quite badly, probably not by the same person who wrote the other parts so far. Grammar is weaker, no explanations are given as paragraphs look like a bunch of sentences put together, and definitions of variables are not clear (even whether they are vectorial or scalar).

Response: Thank you for your appreciation and kind advice. Authors and professional proof-reader in the revised version have now addressed the typos, spelling, and grammatical errors.

  1. Lines 223-224: we don’t get why anxiety detection is a hot investigation by “engineers”.

Response: Thank you for your suggestion. We have revised this sentence and explained as:

Anxiety is a disease that interferes with people of different ages and can last longer, giving an environment for another disease too. Anxiety detection has been accomplished by using SAS questionnaires as a standard technique for a long time [31] rather than detecting changes in human physiology, such as heart rate signals, etc. So far, most of the existing systems are restricted for anxiety detection by observing its biological and biochemical effects, where various techniques, for example, blood pressure and heart rate are physiological effects that could be observed. However, biochemical, physiological, and biological-based methods have shown inconsistent findings because they depend on unstable hormones, depending on several factors such as gender, season, mood, age, well-being status, medicines, and smoking [62 ], [63 ]. On the other hand, anxiety can be detected based on various bio-signals, e.g., EEG, ECG, electromyography, Respiration, Blood Volume Pulses, Blood Pressure, Skin Temperature, and Galvanic Skin Response [64 ], [65], [ 66 ]. Currently, with the growing popularity of the IIoMT, researcher and scientists are more focused on designing wearable sensors that can help to understand human behavior and physiology. So that anxiety disorders can be detected accurately and precisely through IIoMT in a reliable and timely manner.   

  1. Grammar must be checked accurately throughout.

Response: Thank you for your kind advice. The revised manuscript has been revised thoroughly by all the authors to address the typos, spelling, and grammatical errors.

Author Response

Reviewer 2

  1. The title contains “IIoMT” and one statement, which deciphers the abbreviation, contains “IIoMT”, however, an abbreviation IoMT is used through out the manuscript. Consistency is needed in using the abbreviation. I suppose it should be IoMT everywhere.

Response: Thank you for your sharp observations. In the revised manuscript, we have used the terms appropriately and maintained consistency.

The healthcare industry has undergone radical transformations with the integration of artificial intelligence and Internet of medical things (IoMT) as the Intelligent Internet of Medical Things (IIoMT),

  1. The following statement does not sound well; “The main objective of the paper is to detect, estimate and reduce anxiety using yoga, IoMT, and machine learning”. Can the anxiety be reduced using machine learning?

Response: Thank you for helpful suggestions. We have revised the phrase as:

The main objective of the paper is to detect, estimate and reduce anxiety using wearable sensor and SAS, machine learning, and MSY.

  1. “Depression Anxiety Scale-21 (DASS-21)”. The keyword “Stress” is missing in the term definition.

Response: Thank you for your sharp observations. In the revised manuscript, we have rewritten as “Depression Anxiety Stress Scale-21 (DASS-21)”

  1. “designed SaToPillow”. What is a “SaToPillow”?

Response: Thank you for your helpful suggestions. We apologize for the typo mistake. Its a SaYoPillow. We are rewritten the sentence as “Achakonda et al. [ 24] designed SaYoPillow, smart yoga pillow for stress management by taking into consideration of sleeping habits.”

  1. “2.1. smart belt” – typo.

Response: Thank you for your sharp observations. In the revised manuscript, we have corrected the typo.

  1. “W.K. Zuang [27].” – typo in the name.

Response: Thank you for your sharp observations. In the revised manuscript, we have corrected the typo.

  1. No reference to Table 1.

Response: We would like to thank the reviewer for the comments. In the revised manuscript

  1. “For the classification of the data, we implement”. The statement is not complete.

Response: We would like to thank the reviewer for the comments. We have completed the sentence as stated below in the revised manuscript.

For the classification of the data, we implement various intelligent approaches as:

  1. “The DT classifier has the following advantages: it is economical, quick, noise-free, easy to read, and more accurate than other classifiers.” Please provide an evidence that the DT classifier is more accurate than the other classifiers. It is the first. The second, why to use the other classifiers if you know in advance the classifier that is “more accurate than other classifiers”?

Response: We would like to thank the reviewer for the comments. The performance of any classifier depends on various reasons, including dataset used. Even in our results, RF performed better than DT. Hence, we have revised the sentence as “The DT classifier has the following advantages: it is economical, quick, noise-free, and easy to implement than other classifiers”.

  1. “such as random forest, decision tree, and AdaBoost algorithm”. Why the classifier AdaBoost was not introduced and arguments for using it were not provided? Moreover, “algorithm” must be used in plural in the cited statement.

Response: We would like to thank the reviewer for the comments. We have carefully checked the grammatical errors. AdaBoost algorithm is introduced in the revised manuscript, which is as follows:

Adaboost classifier, an ensemble learning method, also called meta-learning, is an iterative approach that incorporates learning from weak classifiers and turning them into strong ones [47-49]. In each iteration, the data samples are trained based on various weights of the classifier to minimize the training errors that can assure accurate predictions. Adaboost must adhere to below requirements [50-51]:

  • On a variety of weighed training instances, the classifier should be trained interactively.
  • It strives to minimize training errors in order to offer a perfect fit for these examples in each iteration.

  1. The metrics like specificity, sensitivity, and others are used in many papers, however, the meaning of TP, TN, FP, and FN usually is not provide

Response: We would like to thank the reviewer for the comments. Here, we have discussed the basic knowledge to understand how various performance metrics such as True Positive (TP), True Negative (TN), False Positive (FP), False Negative (FN), sensitivity, and specificity, are calculated.

Disease Present

Disease Absent

Test Positive

a (TP)

b (FP)

Test Negative

c (FN)

d (TN)

Consider the example of a medical test for diagnosing a condition.

True Positive: Diseased people correctly tested as diseased

False positive: Healthy people incorrectly tested as diseased

True Negative: Healthy people correctly tested as healthy

False Negative: Diseased people incorrectly tested as healthy

Sensitivity refers to the test’s ability to correctly detect diseased patients who do have the condition. Mathematically, this can be expressed as:

Specificity refers to the test’s ability to correctly reject healthy patients without a condition. Mathematically, this can be expressed as:

  1. “So far, most of the existing systems are restricted for anxiety detection by observing its biological and biochemical effects, where various techniques.”. The statement is not complete.

Response: We would like to thank the reviewer for the comments. We have revised and rewritten it.

  1. “However, biochemical, physiological, and biological-based methods have shown inconsistent findings because they depend on unstable hormones, depending on several factors such as gender, season, mood, age, well-being status, medicines, and smoking.”. References are needed for such statement.

Response: We agree with the reviewer. We have cited the statement with the appropriate references.

  1. “On the other hand, anxiety can be detected based on various bio-signals, e.g., EEG, ECG, electromyography, Respiration, Blood Volume Pulses, Blood Pressure, Skin Temperature, and Galvanic Skin Response.”. References are needed for such statement.

Response: We agree with the reviewer. We have cited the statement with the appropriate references.

  1. Units of measurement are not shown in Fig. 3.

Response: Thank you for your sharp observations. In the revised paper, we have included the measurement.

  1. “ad control group respectively.” – two typos.

Response: Thank you for your sharp observations. In the revised manuscript, we have corrected the typo.

  1. “that anxiety and depression can be alleviated by 6 weeks of yoga training.”. It is a quite categorical statement concerning “6 weeks”. May be, 4 weeks is enough. Why did not you investigate? However, may be, it is needed a constant yoga training. May be, the anxiety without yoga returns again. Why, did not you investigate?

Response: Thank you for your sharp observations.

  1. “D.Wedekind’s Pan–Tompkins technique [42].”. Why do you call this technique “D.Wedekind’s Pan–Tompkins technique [42].”? The paper [42] does not contain such a keyword

Response: We acknowledge the reviewer’s correction. It is designed by Pan and Tompkins. We have corrected and updated it.

  1. “D.Wedekind’s”.

Response: Thank you for your sharp observations. In the revised manuscript, we have corrected the typo.

  1. “are better for the classification” can be in comparison with only.

Response: We would like to thank the reviewer for the comments. We have revised this in the updated manuscript.

  1. “(control Vs experimental” – typo.

Response: Thank you for your sharp observations. In the revised manuscript, we have corrected the typo.

  1. “While, in subjects’ classification, CV-3 model of the AB classifier achieved maximum accuracy”. So, you were mistaken (see item 9), when you were writing that the DT classifier is more accurate than other classifiers, were not?

Response: We would like to thank the reviewer for the comments. We have corrected the statement updated manuscript.

  1. “Yoga postures, breathing techniques, and their interconnections with mindfulness-based therapies have dramatically enhanced sleep efficiency caused by anxiety related problems, and cancer sufferers.”. Which reference number is valid for this statement?

Response: We agree with the reviewer. In the updated manuscript, we have added valid referenece.

  1. Figure 7 duplicates results in Table 2. It is not needed.

Response: Thank you for your sharp observations. In the revised manuscript, we have deleted Figure 7.

  1. “We compare the proposed method with the various existing literature”. What is meaning to provide a comparison with the various existing works, if these works were not considered during a review of related work? May be, these works are not related to the research provided?

Response: We agree with the reviewer. Response: Thank you so much for the deep concern, We have compared our study with the previously published work to compare whether our study achieved the maximum accuracy or not, and also support our work.

  1. “The proposed work illustrated an application for detecting, estimating, and reducing anxiety using wearable sensors, SAS questionnaires, machine learning algorithms, and yoga.”. The statement does not look correct. Does it mean that the anxiety was reduced because of using wearable sensors? May be, it is a really true. May be, it is enough to wear sensors and the anxiety will diminish. May be, there is no need for yoga, just sensors!

Response: Thank you for your sharp observations. In the revised manuscript, we have rewritten the sentence as:

The proposed work illustrated an application for detecting, estimating, and reducing anxiety, where wearable sensors and SAS questionnaires were used for detecting the anxiety,  machine learning algorithms for estimating the results obtained from sensors and SAS questionnaires, and yoga is used for reducing the anxiety.

Round 2

Reviewer 1 Report

I suggest the paper be now accepted.

There are only two minor points:

_line 205: the sentence is cut and does not have an end.

_fig. 6 is of a bad quality and needs to be changed

Reviewer 2 Report

Thank you for the revision.